# Net Conversion of Human-Edible Vitamins and Minerals in the U.S. Southern Great Plains Beef Production System

**DOI:** 10.3390/ani12172170

**Published:** 2022-08-24

**Authors:** Phillip A. Lancaster, Deann Presley, Walt Fick, Dustin Pendell, Adam Ahlers, Andrew Ricketts, Minfeng Tang

**Affiliations:** 1Beef Cattle Institute, Kansas State University, Manhattan, KS 66506, USA; 2Department of Agronomy, Kansas State University, Manhattan, KS 66506, USA; 3Department of Agricultural Economics, Kansas State University, Manhattan, KS 66506, USA; 4Department of Horticulture and Natural Resources, Kansas State University, Manhattan, KS 66506, USA

**Keywords:** beef, food system, net contribution, sustainable, upcycling

## Abstract

**Simple Summary:**

Beef production is often viewed as a waste of human-edible food, but overall beef cattle consume primarily feedstuffs nonedible by humans. Previous analyses indicate that the current beef production system is a positive net contributor of high quality protein to the human diet. However, beef provides several important nutrients besides protein: iron, zinc, selenium, phosphorus, vitamin B12, vitamin B6, riboflavin, niacin, and choline. The goal of the current study was to calculate the net nutrient conversion ratio of a beef production system to supply these nutrients to the human diet. The amount of human-absorbable nutrient consumption by beef cattle was calculated as well as the amount of human-absorbable nutrient produced in beef products. The net nutrient conversion ratio was computed as the ratio of nutrient production to nutrient consumption with a value greater than one being a positive net contribution to the human diet. The Southern Great Plains beef production system is a positive net contributor of human-absorbable iron, phosphorus, riboflavin, niacin, and choline to the human diet. Further analysis demonstrates that the amount of corn grain, the primary human-edible feedstuff, consumed by cattle during the feedlot phase is an important indicator of the net nutrient contribution of the beef production system.

**Abstract:**

Beef is a good source of several vitamins and minerals but data on the net contribution to the human diet is lacking. The objective was to quantify the net nutrient contribution of the beef supply chain to provide vitamins and minerals to the human diet. Beef cattle production parameters for the beef supply chain were as described by Baber et al., 2018 with the red and organ meat yield from each production segment estimated using literature values of serially-harvested beef cattle. Nutrient concentration of feeds was acquired from feed composition tables in nutrient requirement texts, and the nutrient concentration of beef and organ meats was based on 2018 USDA Food and Nutrient Database for Dietary Studies. The nutrient absorption coefficients of feeds, red meat, and organs were acquired from the literature. The human-edible conversion ratio was >1.0 for phosphorus when only red meat yield was considered indicating that the beef supply chain produced more human-edible phosphorus than it consumed. When organ meats were included, riboflavin, niacin, choline, and phosphorus had conversion ratios >1.0. After adjusting for the absorption of nutrients, the beef supply chain was a net contributor of niacin and phosphorus in the human diet when accounting for red meat yield only, but when including organ meats, iron, riboflavin, and choline also had conversion ratios >1.0. The maximum proportion of corn in the corn grain plus distillers’ grains component of the feedlot diets for the absorbable conversion ratio to be ≥1 ranged from 8.34 to 100.00% when only red meat yield was considered and from 32.02 to 100.00% when red and organ meats were considered. In conclusion, the current beef production system in the Southern Great Plains produces more human-absorbable iron, phosphorus, riboflavin, niacin, and choline to the human diet than is consumed in the beef supply chain.

## 1. Introduction

Critics have disparaged beef as a food source due to concerns around environmental impacts and consumption of human edible foods. However, beef production systems, even intensive feedlot systems, convert large amounts of human inedible products such as plant biomass into beef, a human edible food. For example, the beef supply chain converts low-quality protein from plant sources into high-quality protein for human consumption [1,2,3]. In addition to protein, beef is a good source of several minerals and vitamins (iron, zinc, selenium, phosphorus, vitamin B12 and B6, riboflavin, niacin, and choline) in the human diet as well as contributing to pet foods.

Recently, Baber et al. [4] reported that the beef supply chain is a net contributor of digestible indispensable amino acids accounting for differences in concentration and digestibility of amino acids in beef compared with plant-based foods. Likewise, vitamins and minerals contained in beef are more available than in plant-based foods [5,6], which can enhance their value in the human diet.

Accurately accounting for the human nutrient supply of the beef production system is necessary to fully assess the sustainability of beef production. Previous analyses have focused on the contribution to human protein supply [2,4,7,8,9], but no analyses have been conducted evaluating other nutrients with high concentrations in beef. Therefore, the objective of this study is to determine the net nutrient contribution of the beef supply chain as a mineral and vitamin source to the human diet.

## 2. Materials and Methods

No animals were used in this research and Institutional Animal Care and Use Committee approval was not required.

A summative model of net nutrient contribution (NNC) was developed based on the current industry diets (Table 1) and production parameters (Table 2) reported by Baber et al. [4] for the entire beef supply chain. This was done so that results would align with the net protein contribution reported by Baber et al. [4]. The cow-calf phase included a breeding population to produce calves for slaughter. Calves from the cow-calf phase moved into the stocker, then feedlot phase where the cow-calf production cycle represents 365 days, and the stocker and feedlot phases represent the time animals were managed in those sectors of the industry. A portion of calves (22.8%) from the cow-calf phase moved directly to the feedlot phase to represent current industry practices. Additionally, open replacement heifers were moved directly to the feedlot phase.

Production parameters adapted from Baber et al. [4].

Model diets were based on typical US beef cattle feedstuffs used in each phase of the Southern Great Plains beef production system. Cows and nursing calves consumed pasture along with small amount of protein supplement as cottonseed meal. In the stocker phase, calves consumed wheat forage and small amount of corn grain and distiller’s grains. The feedlot phase was divided into a receiving phase using a typical receiving diet in Southern Great Plains feedlots, and a finishing phase using a typical finishing diet for Southern Great Plains feedlots. 

Human-edible nutrient produced was computed for each production phase and the entire beef supply chain. Red and organ meat yield was estimated from published serial harvest studies [10,11,12,13,14,15,16] and represented the change in weight that occurs in each production phase (Appendix A). Estimation of the nutrient content of feeds, beef meat, and beef organ meats (liver, heart, kidney, spleen, pancreas, gastrointestinal tract) were gathered from nutrient composition tables and published literature (Appendix A). If data were not available, as sometimes happened with organ meats, that organ was not included in the analysis. Values used for corn silage were same as those for corn grain assuming that corn silage is 50% corn grain, and that the corn grain would be 100% human-edible if harvested as grain rather plant biomass. The amount of human-edible nutrient produced was based on that amount of animal product produced in each phase such that each phase was independent rather than a running total. The amount of human-edible nutrient consumed and produced for the entire supply chain was the sum of the three production phases. Human-edible conversion ratio was then computed as the amount of human-edible nutrient produced in beef products to the amount of human-edible nutrient consumed in feed, thus a value greater than 1 indicates that the supply chain is a net contributor to the human diet. 

Human-absorbable nutrient consumed and produced was computed for each production phase and the entire beef supply chain by multiplying the human-edible nutrient consumed and produced by the nutrient absorption coefficient. Estimates of nutrient absorption coefficients from feed, beef meat, and beef organ meats were gathered from published studies using humans, swine or poultry (Appendix A). Swine have been an acceptable model for assessing nutrient digestibility in humans [17]. If a feedstuff had a human-edible fraction of zero, then the human absorption coefficient was assumed to be zero. Studies on the digestibility of nutrients from organ meats other than liver could not be found in a literature search, thus the nutrient absorption coefficient value for liver was used on the basis that in vitro protein digestibility of heart, kidney, and spleen are similar to liver [18]. The amount of human-absorbable nutrient produced was based on that amount of animal product produced in each phase such that each phase was independent rather than a running total. The amount of human-absorbable nutrient consumed and produced for the entire supply chain was the sum of the three production phases. The human-absorbable conversion ratio was then computed as the amount of human-absorbable nutrient produced in beef products to the amount of human-absorbable nutrient consumed in feed; thus a value greater than 1 indicates that the supply chain is a net contributor to the human diet.

Human-edible and human-absorbable conversion ratios were computed for iron (Fe), zinc (Zn), selenium (Se), phosphorus (P), vitamin B6 (B6), riboflavin, niacin, niacin + tryptophan, and choline. Tryptophan is a precursor for niacin synthesis [19] with a conversion efficiency of 60 mg of tryptophan producing 1 mg of niacin [20]. The human-edible and human-absorbable conversion ratios for niacin were computed with and without inclusion of tryptophan. 

The primary human-edible feedstuff consumed by cattle is corn grain; thus, the amount of corn grain fed to cattle is the primary factor by which the beef industry can affect the net nutrient conversion ratio. Feedlot diets are the main source of corn fed to beef cattle where corn can be replaced with corn byproducts. The proportion of corn in the corn grain plus distillers’ grains component of the feedlot diets was varied from 0 to 100% by increments of 10, and the resulting net nutrient conversion ratios for red meat and red plus organ meats were recorded. The relationship between the proportion of corn and the net nutrient conversion ratio was curvilinear. A non-linear model was fit to the data using Origin software (ver. 2022b; OriginLab, Northampton, MA, USA; https://www.originlab.com/; Accessed on 1 May 2022). The best fit based on adjusted coefficient of determination and the Chi-square was a two-phase exponential decay function.
Y = A1 × e(−x/t1) + A2 × e(−x/t2) + y0(1)
where, Y is the net nutrient conversion ratio, A1 and A2 are the two time constants, t1 and t2 are the two rate constants, y0 is the y-intercept, and x is the proportion of corn in the corn grain plus distillers’ grains component of the feedlot diets. The GoalSeek function in Microsoft Excel was then used to find the proportion of corn at which the net nutrient conversion ratio is equal to or greater than 1.

## 3. Results

Among the beef production phases, human-edible nutrient consumption was greatest for all nutrients in the feedlot phase as is expected, as corn grain is the primary human-edible feedstuff used in beef cattle diets and the majority of corn grain is consumed in the feedlot phase of production (Table 3). Human-edible nutrient production accounting for red meat yield only was least for the stocker phase, and approximately equal between the cow-calf and feedlot phases for all nutrients. The cow-calf phase had the greatest human-edible nutrient conversion ratio for all investigated mineral and B-vitamin nutrients except Zn, Se, and P. However, only P had a positive human-edible net contribution to the human diet in the entire beef supply chain when considering red meat consumption only. The beef supply chain has a large positive contribution of vitamin B12 to the human diet as B12 is not produced in plants, and thus there is not a tradeoff between human-edible feedstuffs and beef. 

When organ meats were included in the computation of human-edible nutrient production, the pattern was similar among production phases as when only red meat yield was used (Table 4). The net nutrient conversion ratios were increased compared to consideration of only red meat consumption, resulting in a net positive contribution for riboflavin, niacin, niacin plus tryptophan, and choline, along with P, for the entire beef supply chain. Phosphorus is the only nutrient of those evaluated with a net positive contribution to the human diet in the feedlot phase when both red meat and organ meat were considered.

After accounting for differences in absorption between human-edible feedstuffs and red meat, the beef supply chain had net positive contribution of P, niacin, and niacin plus tryptophan to the human diet (Table 5). The feedlot phase has the greatest human-absorbable nutrient consumption whereas the cow-calf phase has the greatest nutrient conversion ratio for all nutrients except for Zn and P. The feedlot phase had the greatest human-absorbable nutrient production for all nutrients, and had the lowest nutrient conversion ratios for all nutrients except for Zn and P. The positive net contribution of niacin and niacin plus tryptophan is somewhat different than the results for human-edible nutrient conversion ratios in that accounting for absorption resulted in conversion ratios greater than one.

With the inclusion of organ meats in the calculation of human-absorbable nutrient produced, the beef supply chain was a net contributor of Fe, P, riboflavin, niacin, niacin plus tryptophan, and choline to the human diet compared to only P, niacin, and niacin plus tryptophan when only red meat yield was included (Table 6). Similar to when only red meat was used in the calculation of human-absorbable nutrient produced, the stocker phase had the least human-absorbable nutrient produced for all nutrients, and the feedlot phase had the lowest nutrient conversion ratios for all nutrients except for Zn. The cow-calf phase had the greatest human-absorbable nutrient conversion ratios for all nutrients except for Zn.

The proportion of corn in the corn grain plus distillers’ grains component of the feedlot diet where the net nutrient conversion ratio is equal to or greater than one is presented in Table 7. When organ meats were included in the calculation of human-edible or human-absorbable nutrient produced, the proportion of corn that could be used in feedlot diets increased. For P, 100% corn could be used in the corn grain plus distillers’ grains component of feedlot diets regardless of whether evaluating human-edible or human-absorbable nutrient conversion ratio with or without organ meats. For Zn and Se, there was no proportion of corn that would result in a net nutrient conversion ratio equal to or greater than one. When evaluating human-edible net nutrient conversion ratio, the maximum proportion of corn that could be used in the corn grain plus distillers’ grains component of feedlot diets was 0.00% for red meat yield only and 5.65% for red and organ meat yield with Fe being the limiting nutrient in both cases. However, based on human-absorbable net nutrient conversion ratio, the maximum proportion of corn that could be used in the corn grain plus distillers’ grains component of feedlot diets was 9.57% for red meat yield only and 32.15% for red and organ meat yield with vitamin B6 being the limiting nutrient in both cases.

## 4. Discussion

Beef is a good source of many vitamins and minerals for humans; iron, zinc, selenium, phosphorus, vitamin B6 (pyridoxine), vitamin B12 (cobalamin), riboflavin, niacin, and choline [6,21]. Iron is a key component of hemoglobin and deficiency can result in anemia especially in women of child-bearing age [22]. Zinc is a component of many enzymes in the body and deficiency can impair immune function and reproduction. Selenium is an important component of glutathione peroxidase mitigating oxidative damage to cells. Phosphorus in the form of phosphate is an important component of bone structure and a key part of energy metabolism as are riboflavin and niacin [20]. Phosphorus deficiency can lead to abnormal bone growth (ricketts) and osteomalacia. Vitamin B6 and B12 are essential to amino acid metabolism, and B12 is an important component in folate metabolism and nucleic acid synthesis. Additionally, absorption of vitamins and minerals from beef by humans is greater than that for plant sources [5,6,22].

Vitamin B12 synthesis is limited almost exclusively to microorganisms, and the vitamin is only present in animal food products [20]. Meat and liver are excellent sources of the vitamin with beef meat and liver having appreciably greater concentrations than pork or chicken due to the synthesis of B12 by rumen microorganisms. The lack of vitamin B12 in plants indicates that the beef production system consumes no human-edible B12 resulting in a positive net contribution to the human diet; however, since the input of human-edible B12 to the beef production system is zero, a net nutrient conversion ratio cannot be computed.

To our knowledge no previous reports have published the net vitamin and mineral contribution of beef production systems to the human diet, but several previous analyses of net protein conversion have been published. Early estimates of protein conversion efficiency indicated that beef required 9 to 19 kg of protein to produce 1 kg of edible meat protein, which was 30 to 900% greater than eggs, poultry, pork or milk, due to using total rather than human-edible protein intake [23,24,25]. Ertl et al. [26] indicated that human-edible protein conversion efficiency averaged 1.52 for beef cattle in Austria, which was greater than for pork, eggs, poultry, and mutton, but not milk. Additionally, accounting for the protein quality or biological value of plant vs. meat protein increased the net protein conversion efficiency of beef from 1.52 to 2.81 which compares more favorably with 3.78 (milk), 0.56 to 0.76 (poultry and pork), and 1.04 (eggs and mutton) for other livestock products [26]. Similarly, for many vitamins and minerals absorption by humans is greater from beef than from plant sources [5,6,22]. Adjusting the net nutrient conversion ratio for differences in nutrient absorption between human-edible feedstuffs and beef increased the net nutrient conversion ratio of several nutrients in the current analysis and resulted in the beef production system having a positive net contribution for additional nutrients.

Organ meats are excellent sources of many vitamins and minerals. The organ meats used in this analysis included heart, liver, kidney, spleen, pancreas, and gastrointestinal tract; all of which could be consumed by humans. Including organ meats in the output of human-edible and human-absorbable nutrients increased the net nutrient conversion ratio for all nutrients, and resulted in positive net contributions for Fe, riboflavin, niacin, and choline, but rarely are these organ meats consumed by the US population. Approximately 1.36 million metric tons of organ meats are produced in the beef supply chain annually (calculated from USDA AMS statistics). The pet food industry utilizes 136,000 metric tons of organ meats annually [27] and 300,000 metric tons are exported annually (U.S. Meat Export Federation). It is unlikely that the US population consumes the remaining 924,000 metric tons of organ meats produced (consumption data is unavailable) indicating that most of the organ meats are used for non-food purposes. Consumption of nutrients by pets is not normally included in the analysis of net contribution of nutrients from the beef production system, but many household pets being monogastric animals as humans are benefit from more bioavailable nutrients in beef. Consumption of pet edible feedstuffs such as corn by the beef production system has the same implications as for the human diet as it is using land to produce animal feed rather than directly producing pet food. Increasing the consumption of organ meats by humans and pets would improve the nutrient conversion efficiency of beef production.

Similar to the differences in mineral and B-vitamin nutrient conversion ratios among production phases in the current analysis, Baber et al. [4] reported that the feedlot phase of production consumed the most human-edible protein, whereas the cow-calf phase had the greatest human-edible and net protein conversion efficiency. This trend is based on the diet ingredients used in the different sectors of the beef industry where the vast majority of feedstuffs used in the cow-calf phase are non-edible by humans compared with the feedlot phase where approximately 50% of the feedstuffs are edible by humans, primarily corn grain. Grass-finished beef production systems utilize almost exclusively feedstuffs non-edible by humans resulting in net protein contribution 800 times greater (1597 vs. 1.96) than grain-finished production systems [28]. Additionally, the human-edible protein conversion ratio was 6.1 for Argentina beef production compared with 1.19 for US beef production due to the fact that cattle in Argentina mostly consume pasture and byproducts non-edible by humans [25]. Thus, the net nutrient contribution of any beef production system is primarily determined by the amount of human-edible feedstuffs used in the different production systems.

In agreement with analysis of Thomas et al. [28] and Broderick [25], the analysis of the proportion of corn in the corn grain plus distillers’ grains component of the feedlot diet indicated that corn consumption is an important component of the net nutrient contribution of the beef production system to the human diet. With the exception of Zn and Se, there is a proportion of corn that will allow for a positive net nutrient contribution for all nutrients. However, the proportion of corn required for a positive net contribution for vitamin B6 is low and may not be very practical as inclusion of wet distillers’ grains above 40% of diet dry matter reduces cattle performance [29,30]. To maintain a maximum of 40% wet distillers’ grains in the feedlot diet, the minimum proportion of corn in the corn grain plus distillers’ grains component of the feedlot diet would be 43%. A proportion of 43% would not allow a positive net contribution of iron, B6, riboflavin, or choline when using red meat only, but when organ meats are added, the beef production system becomes a positive net contributor of iron, riboflavin, niacin, and choline at a proportion of 43% corn. The reason for the poor net nutrient conversion ratios for Zn and Se and the lack of a proportion of corn that will allow a positive net contribution is that a large amount of each nutrient is consumed in mineral supplements in forms that are human edible, and that the absorption coefficients for beef are more similar to human-edible feedstuffs than for other nutrients (Appendix A).

With corn being the primary human-edible feedstuff consumed by beef cattle, the nutrient concentration and availability in corn is expected to be important to the net nutrient conversion ratio. Ertl et al. [8] suggested that animal production systems could be evaluated based on their ability to transform human-edible protein inputs to animal protein by multiplying the ratio (output/input) of protein quantity with the ratio (output/input) of protein quality. Based on this concept, the ratio of nutrients of beef to corn is likely a good indicator of the net nutrient conversion ratio. The ratio of nutrient concentration in beef to corn multiplied by the ratio of absorption coefficients in beef to corn for each nutrient was strongly correlated (r = 0.99) with the net nutrient conversion ratio.

## 5. Conclusions

The cow-calf phase of the beef supply chain consumes the least amount of human-edible mineral and B-vitamin nutrients for most nutrients evaluated, resulting in the greatest net nutrient contribution to the human diet. The feedlot phase has the lowest net nutrient contribution to the human diet for all evaluated mineral and B-vitamin nutrients except phosphorus. The stocker phase generally had the least human-edible nutrient consumption and production with net nutrient contribution intermediate of the cow-calf and feedlot phases. Adjusting nutrient consumption and production based on absorbable nutrient improved the net nutrient contribution of the beef supply chain to the human diet as did inclusion of organ meats in the nutrient production calculation. The beef supply chain is a net positive contributor of iron, phosphorus, vitamin B12, riboflavin, niacin, and choline to the human diet. The amount of corn grain consumed by cattle is a primary determinant of the net nutrient contribution of beef production systems. The net nutrient contribution could be improved by decreasing as much as possible the proportion of human-edible feedstuffs in cattle diets and utilizing as much organ meats as possible in human and pet diets.

## Figures and Tables

**Table 1 animals-12-02170-t001:** Ingredient composition and human-edible fraction of feedstuffs for each phase of the beef supply chain used in the summative model.

Production Phase and Diet Ingredient	Human-Edible Fraction, % ^1^	Dietary Amount, % as-Fed
Cow-calf phase		
Bermudagrass, fresh	0	97.75
Cottonseed meal	0	1.99
Corn grain (filler in mineral)	100	0.01
Mineral supplement	Nutrient dependent ^2^	0.25
Stocker phase		
Wheat forage	0	97.12
Corn grain	100	1.00
Distiller’s grains, dry	0	1.50
Mineral supplement	Nutrient dependent ^2^	0.38
Feedlot receiving phase		
Alfalfa hay	0	16.70
Corn silage	50	26.36
Steam-flaked corn	100	18.01
Distiller’s grains, wet	0	35.24
Molasses	100	1.76
Urea	0	0.53
Mineral supplement	Nutrient dependent ^2^	1.40
Feedlot finishing phase		
Alfalfa hay	0	2.12
Corn silage	50	20.43
Steam-flaked corn	100	42.24
Distiller’s grains, wet	0	29.58
Molasses	100	2.79
Urea	0	0.72
Tallow	0	0.62
Mineral supplement	Nutrient dependent ^2^	1.49

^1^ Percent of feed ingredient that is human-edible. ^2^ Iron = 0% as iron oxide, Zinc = 100% as zinc sulfate, Selenium = 100% as selenite, Phosphorus = 100% mono- or di-calcium phosphate, vitamins (B12, B6, riboflavin, niacin, choline) = not applicable.

**Table 2 animals-12-02170-t002:** Production parameters used in the summative model to estimate human edible nutrient intake, production, and conversion ratio.

Parameter	Value
Cow-calf phase	
Days on feed, d	365
Age of calf at weaning, d	207
Cows per bull	24
Calving rate, %	88.6
Calf mortality rate, %	4.0
Cow mortality rate, %	2.8
Cow culling rate, %	10.2
Calves sent direct to feedlot, %	22.8
Calves sent to stocker, %	77.2
Replacement heifers per cow	0.19
Mature cow body weight, kg	571
Bull body weight, kg	907
Weaned steer body weight, kg	253
Weaned heifer body weight, kg	240
Replacement heifer body weight at breeding, kg	342
Dry cow feed intake, kg DM/d	10.40
Lactating cow feed intake, kg DM/d	12.75
Bull feed intake, kg DM/d	18.45
Replacement heifer feed intake, kg DM/d	7.20
Steer calf feed intake, kg DM/d	3.42
Heifer calf feed intake, kg DM/d	3.28
Stocker phase	
Days on feed, d	129
Mortality rate, %	1.5
Steer body weight entering feedlot, kg	360
Heifer body weight entering feedlot, kg	326
Steer feed intake, kg DM/d	6.92
Heifer feed intake, kg DM/d	6.52
Feedlot phase	
Days on feed, d	159
Mortality rate (heavyweight), %	1.3
Mortality rate (lightweight), %	2.0
Finished steer body weight, kg	649
Finished heifer body weight, kg	588
Steer feed intake, kg DM/d	10.17
Heifer feed intake, kg DM/d	9.21

**Table 3 animals-12-02170-t003:** Human-edible nutrient conversion efficiency of the beef supply chain (red meat yield only).

Item	Iron	Zinc	Selenium	Phosphorus	B6	Riboflavin	Niacin	Niacin + Trp	Choline
Cow-calf									
Intake, mg	7641	15,475,832	78,531	241,130	1161	183	4126	5654	107,277
Production, mg	228,463	519,313	2364	1,852,305	36,905	19,683	448,138	825,980	7,987,407
Conversion ratio	29.90	0.03	0.03	7.68	31.78	107.33	108.61	146.08	74.46
Stocker									
Intake, mg	90,604	2,244,941	17,760	40,115	13,772	2174	48,926	67,047	1,272,080
Production, mg	58,440	132,840	605	473,817	9440	5035	114,633	211,285	2,043,169
Conversion ratio	0.65	0.06	0.03	11.81	0.69	2.32	2.34	3.15	1.61
Feedlot									
Intake, mg	2,692,911	8,557,328	65,609	221,400	442,682	77,254	1,508,601	2,104,547	38,994,033
Production, mg	246,866	561,147	2554	2,001,518	39,878	21,268	484,238	892,517	8,630,833
Conversion ratio	0.09	0.07	0.04	9.04	0.09	0.28	0.32	0.42	0.22
Supply chain									
Intake, mg	2,791,156	26,278,101	161,901	502,644	457,615	79,611	1,561,654	2,177,248	40,373,391
Production, mg	533,770	1,213,299	5522	4,327,640	86,224	45,986	1,047,010	1,929,783	18,661,409
Conversion ratio	0.19	0.05	0.03	8.61	0.19	0.58	0.67	0.89	0.46

**Table 4 animals-12-02170-t004:** Human-edible nutrient conversion efficiency of the beef supply chain (red + organ meat yield).

Item	Iron	Zinc	Selenium	Phosphorus	B6	Riboflavin	Niacin	Niacin + Trp	Choline
Cow-calf									
Intake, mg	7641	15,475,832	78,531	241,130	1161	183	4126	5654	107,277
Production, mg	544,957	544,247	5054	4,086,275	77,290	52,137	982,921	1,760,136	17,687,328
Conversion ratio	71.32	0.04	0.06	16.95	66.55	284.31	238.22	331.30	164.87
Stocker									
Intake, mg	90,604	2,244,941	17,760	40,115	13,772	2174	48,926	67,047	1,272,080
Production, mg	140,755	138,438	1307	1,050,408	20,058	14,242	255,141	455,268	4,649,087
Conversion ratio	1.55	0.06	0.07	26.18	1.46	6.55	5.21	6.79	3.65
Feedlot									
Intake, mg	2,692,911	8,557,328	65,609	221,400	442,682	77,254	1,508,601	2,104,547	38,994,033
Production, mg	568,475	580,454	5357	4,338,670	83,251	54,995	1,049,806	1,886,950	18,925,592
Conversion ratio	0.21	0.07	0.08	19.60	0.19	0.70	0.70	0.90	0.49
Supply chain									
Intake, mg	2,791,156	26,278,101	161,901	502,644	457,615	79,611	1,561,654	2,177,248	40,373,391
Production, mg	1,254,188	1,263,138	11,717	9,475,353	180,600	121,374	2,287,868	4,102,353	41,262,008
Conversion ratio	0.45	0.05	0.07	18.85	0.39	1.52	1.47	1.88	1.02

**Table 5 animals-12-02170-t005:** Human-absorbable nutrient conversion efficiency of the beef supply chain (red meat yield only).

Item	Iron	Zinc	Selenium	Phosphorus	B6	Riboflavin	Niacin	Niacin + Trp	Choline
Cow-calf									
Intake, mg	458	1,733,624	42,431	204,708	621	115	1650	2873	80,458
Production, mg	41,417	188,698	2104	1,278,091	32,846	17,518	378,677	747,073	7,588,036
Conversion ratio	90.34	0.11	0.05	6.24	52.86	151.63	229.44	260.04	94.31
Stocker									
Intake, mg	5436	255,352	9878	31,102	7368	1370	19,570	34,067	954,060
Production, mg	10,594	48,269	538	326,934	8402	4481	96,865	191,100	1,941,010
Conversion ratio	1.95	0.19	0.05	10.51	1.14	3.27	4.95	5.61	2.03
Feedlot									
Intake, mg	154,026	1,078,861	49,017	107,492	242,450	48,670	620,661	1,100,288	29,147,368
Production, mg	44,753	203,898	2273	1,381,047	35,492	18,929	409,181	807,253	8,199,292
Conversion ratio	0.29	0.19	0.05	12.85	0.15	0.39	0.66	0.73	0.28
Supply chain									
Intake, mg	159,921	3,067,837	101,327	343,302	250,440	50,155	641,882	1,137,228	30,181,887
Production, mg	96,764	440,864	4915	2,986,072	76,740	40,928	884,723	1,745,427	17,728,338
Conversion ratio	0.61	0.14	0.05	8.70	0.31	0.82	1.38	1.53	0.59

**Table 6 animals-12-02170-t006:** Human-absorbable nutrient conversion efficiency of the beef supply chain (red + organ meat yield).

Item	Iron	Zinc	Selenium	Phosphorus	B6	Riboflavin	Niacin	Niacin + Trp	Choline
Cow-calf									
Intake, mg	458	1,733,624	42,431	204,708	621	115	1650	2873	80,458
Production, mg	92,516	383,902	4498	2,806,172	68,788	47,168	843,998	1,601,459	16,683,085
Conversion ratio	201.80	0.22	0.11	13.71	110.71	408.28	511.38	557.43	207.35
Stocker									
Intake, mg	5436	255,352	9878	31,102	7368	1370	19,570	34,067	954,060
Production, mg	23,815	98,089	1163	721,184	17,852	12,925	219,605	414,626	4,377,241
Conversion ratio	4.38	0.38	0.12	23.19	2.42	9.44	11.22	12.17	4.59
Feedlot									
Intake, mg	154,026	1,078,861	49,017	107,492	242,450	48,670	620,661	1,100,288	29,147,368
Production, mg	97,727	412,818	4768	2,981,935	74,094	49,693	899,692	1,715,598	17,862,838
Conversion ratio	0.63	0.38	0.10	27.74	0.31	1.02	1.45	1.56	0.61
Supply chain									
Intake, mg	159,921	3,067,837	101,327	343,302	250,440	50,155	641,882	1,137,228	30,181,887
Production, mg	214,058	894,000	10,428	6,509,291	160,734	109,787	1,963,295	3,731,684	38,923,164
Conversion ratio	1.34	0.29	0.10	18.96	0.64	2.19	3.06	3.28	1.29

**Table 7 animals-12-02170-t007:** Proportion of corn grain replacing distiller’s grains in feedlot diets where net nutrient conversion is equal to or greater than 1.

Item	Iron	Zinc	Selenium	Phosphorus	B6	Riboflavin	Niacin	Niacin + Trp	Choline
Red meat only									
Edible conversion	NP	NP	NP	100.00	0.64	25.43	35.54	47.19	21.53
Absorbable conversion	21.98	NP	NP	100.00	9.57	43.62	81.16	89.52	30.48
Red and organ meat									
Edible conversion	5.65	NP	NP	100.00	15.33	95.42	86.13	100.00	58.25
Absorbable conversion	86.27	NP	NP	100.00	32.15	100.00	100.00	100.00	74.43

NP = not possible, proportion of corn necessary for net nutrient conversion equal to one is below 0%.

## Data Availability

Data are available upon request.

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
