# Peer review of "Net Conversion of Human-Edible Vitamins and Minerals in the U.S. Southern Great Plains Beef Production System"

_animals, 2022, doi:10.3390/ani12172170_

Round 1

Reviewer 1 Report

An interesting study I enjoyed reading. 

Lines 96-97 - exactly how were the contributions from the total supply chain calculated? I can make an educated guess but, ideally, it should be explicit. 

Lines 116-118 - what is the support/justification for defaulting to liver absorption coefficients for organs where there are no values? Has there been any work confirming that they are relevant. If not this needs to be highlighted as using liver values could skew the results.  

Line 225 - If not possible to calculate B12 nutrient conversion ratio, what about looking at Co instead?

Table S2 - values are DM for feedstuffs and wet weight for meat and organs. I realise people generally eat 'wet' weight (but then so do the animals) but for strict comparison should it not be either both DM or both wet? 

Author Response

Reviewer 1

Comments and Suggestions for Authors

An interesting study I enjoyed reading. 

Lines 96-97 - exactly how were the contributions from the total supply chain calculated? I can make an educated guess but, ideally, it should be explicit. 

Response: the total supply chain was the sum of all three phases. This has been added to the manuscript.

Lines 116-118 - what is the support/justification for defaulting to liver absorption coefficients for organs where there are no values? Has there been any work confirming that they are relevant. If not this needs to be highlighted as using liver values could skew the results.

Response: we based this off of a study reporting that protein digestibility is similar between liver and other organ meats and the fact that the vitamins and mineral would be in the same form in these organ meats. There is a reasonable possibility that this is not entirely accurate, but there is no data on vitamin and mineral digestibility in these organ meats that we can find. If the reviewer knows of such data please let us know and we will revise.  

Line 225 - If not possible to calculate B12 nutrient conversion ratio, what about looking at Co instead?

Response: We did the calculations with cobalt, but again there is a problem. Inorganic forms of cobalt are edible by humans, but humans have no requirement for cobalt except in the form of B12 and no ability to absorb any B12 synthesized by bacteria in the colon. Thus, even though we can calculate a conversion ratio (Co to B12), it does not make biological sense. Additionally, the ratio is very near 0 because the beef production system consumes a large amount of cobalt in trace mineral and the conversion of Co to B12 is only 1-3%.

Table S2 - values are DM for feedstuffs and wet weight for meat and organs. I realise people generally eat 'wet' weight (but then so do the animals) but for strict comparison should it not be either both DM or both wet? 

Response: Whether on a DM or wet basis the results would be the same. For the calculations, we would adjust the nutrient concentration in the feed or meat, and the total amount (mg) consumed or produced would be the same.

Reviewer 2 Report

The title seems misleading since that paper is focused on vitamins and minerals, not all of the net nutrients.  Maybe consider rewording the title.

This article addresses the net nutrient conversion for vitamins and minerals; however, the initial data used for the calculation is based on a small region of the US.  This reviewer believes that the methodology in sound; however, to make the broad statement for the US beef supply chain when only one production system representative, this reviewer cannot support this article.  However, if the title and reference to the specific region of the country that the assumptions are based on was changed, I think this would be an acceptable paper. 

Line 83: Table 1 – underline phases to assist readers to make easier to follow.  Additionally, nutrient dependent – I’m not 100% sure; however, I believe that each nutrient dependent should have a superscript 2.

Table 1: diet ingredients does not appear to represent the US, these examples of feeds especially for the cow-calf and stocker phase represent the southern plains. 

·         On the cow-calf section; would it represent the entire US if you used wording such as fresh forage instead of Bermudagrass?  Additionally, using protein supplement instead of cottonseed meal.  How did you account for the large portion of cows that are supplemented with harvested feeds in the winter period that could be hay, haylage, corn silage, grain or etc? 

·         Stocker phase – you have wheat forage, does that represent the calves in a stocker or backgrounding phase? 

·         Feedlot phase – is steam-flaked corn an accurate representation of the type of corn used in the feedlot phase?  Or would corn grain without the processing method be a better way to describe the primary starch source? 

Line 87: Table 2

·         Underline each of the phases to help reader follow the information easier.

·         Many of the values seem low, they appear to represent the southern plains; however, if this article is trying to represent the US values should include all region of the US.  This reviewer understands that you are taking your numbers from Baber et al. paper but that papers seems to have the same problem of a limited region of the US.

Line 91: Authors talk about cows and calves consuming pasture, how about the cows that needs to consume harvested feed in the winter?  A large portion of the cattle in the US would be in a region where winter feeding is required.  This paper does not appear to address this issue.

Line 92: Wheat forage is only used in a small portion of the US.  If this paper is address US beef production, why are you only considering wheat forage.  This reviewer understands that the wheat forage would be similar to other forages provided to the stockers/backgrounder however, using wheat forage appears to limit the value of this paper. 

Table 3, 4, 5, 6, & 7: Underlining the phase or grouping makes it easier for the reader to follow quickly.

Author Response

Reviewer 2

Comments and Suggestions for Authors

The title seems misleading since that paper is focused on vitamins and minerals, not all of the net nutrients.  Maybe consider rewording the title.

Response: The title was changed to focus on vitamins and minerals.

This article addresses the net nutrient conversion for vitamins and minerals; however, the initial data used for the calculation is based on a small region of the US.  This reviewer believes that the methodology in sound; however, to make the broad statement for the US beef supply chain when only one production system representative, this reviewer cannot support this article.  However, if the title and reference to the specific region of the country that the assumptions are based on was changed, I think this would be an acceptable paper. 

Response: We agree that analysis does not include all possible beef cattle diets used across the US. Corn is the primary feedstuff that would differ. The cow-calf phase generally represents the entire US except for the corn belt, and the feedlot phase generally represents the entire US except the northwest. The main difference would be the stocker phase where backgrounding operations in drylot would feed more corn compared to the grazing operations in the southern plains. We chose to use these diets to match previous analysis (Baber et al., 2018), and the data are not available to allocate the proportion of total US cattle in different production systems. In accordance with this suggestion we have changed to focus on the southern plains production system.

Line 83: Table 1 – underline phases to assist readers to make easier to follow.  Additionally, nutrient dependent – I’m not 100% sure; however, I believe that each nutrient dependent should have a superscript 2.

Response: corrected.

Table 1: diet ingredients does not appear to represent the US, these examples of feeds especially for the cow-calf and stocker phase represent the southern plains. 

Response: the description of the production system has been changed to the southern plains rather than the US

  • On the cow-calf section; would it represent the entire US if you used wording such as fresh forage instead of Bermudagrass?  Additionally, using protein supplement instead of cottonseed meal.  How did you account for the large portion of cows that are supplemented with harvested feeds in the winter period that could be hay, haylage, corn silage, grain or etc? 

      Response: As stated above, the cow-calf phase is generally representative of the US cow-calf phase except in the corn belt where more corn grain and corn silage would be fed; hay and haylage would not change the results as they are not human edible. We have adjusted the focus of the paper on the southern plains production system.

  • Stocker phase – you have wheat forage, does that represent the calves in a stocker or backgrounding phase? 

Response: the stocker and backgrounding phase are different terms for generally the same phase of production. The term stocker is used more in the southern US indicating grazing and backgrounding in the northern US indicating drylot.

  • Feedlot phase – is steam-flaked corn an accurate representation of the type of corn used in the feedlot phase?  Or would corn grain without the processing method be a better way to describe the primary starch source? 

Response: For the purposes of this analysis, the term corn grain could be used because processing does not impact the vitamin or mineral concentration.

Line 87: Table 2

  • Underline each of the phases to help reader follow the information easier.

Response: corrected.

  • Many of the values seem low, they appear to represent the southern plains; however, if this article is trying to represent the US values should include all region of the US.  This reviewer understands that you are taking your numbers from Baber et al. paper but that papers seems to have the same problem of a limited region of the US.

Response: We have adjusted the paper to focus on the southern plains. Data are not available to easily compute the proportion of total US cattle fed different diets.

Line 91: Authors talk about cows and calves consuming pasture, how about the cows that needs to consume harvested feed in the winter?  A large portion of the cattle in the US would be in a region where winter feeding is required.  This paper does not appear to address this issue.

Response: the majority of harvested feed used in the winter would be forage with possibly some human-edible grain supplementation, but mostly human-inedible grain byproducts (distillers grains, wheat midds, soybean hulls, etc.). In the southern plains, the winter ration would be free-choice hay with protein supplement.

Line 92: Wheat forage is only used in a small portion of the US.  If this paper is address US beef production, why are you only considering wheat forage.  This reviewer understands that the wheat forage would be similar to other forages provided to the stockers/backgrounder however, using wheat forage appears to limit the value of this paper. 

Response: the land area of wheat grazing is relatively small, but a very large percentage of calves from all across the US are shipped to the southern plains to graze winter wheat pasture or summer pasture during the stocker phase. This production system represents a large proportion of animals.

Table 3, 4, 5, 6, & 7: Underlining the phase or grouping makes it easier for the reader to follow quickly.

Response: corrected.

Round 2

Reviewer 2 Report

No suggestions for changes